Distinctive courtship phenotype of the Vogelkop Superb Bird-of-Paradise Lophorina niedda Mayr, 1930 confirms new species status

Scholes Edwin es269@cornell.edu edwin.scholes@cornell.edu 1
Laman Timothy G. 2
1 Cornell Lab of Ornithology, Cornell University , Ithaca , NY , United States of America
2 Museum of Comparative Zoology, Harvard University , Cambridge , MA , United States of America
Edwards Scott
Electronic publication date: 2018 Apr 16
Publication date: 2018
Volume: 6
Electronic Location ID: e4621
Received 2018 Feb 16; Accepted 2018 Mar 24
Copyright: ©2018 Scholes and Laman
Copyright year: 2018
Copyright holder: Scholes and Laman
License: This is an open access article distributed under the terms of the Creative Commons Attribution License, which permits unrestricted use, distribution, reproduction and adaptation in any medium and for any purpose provided that it is properly attributed. For attribution, the original author(s), title, publication source (PeerJ) and either DOI or URL of the article must be cited.
License URL: https://creativecommons.org/licenses/by/4.0/

Keywords: Display behavior, Paradisaeidae, New Guinea, Video analysis, Birds-of-paradise

Funding: Cornell Lab of Ornithology This work was supported by the Cornell Lab of Ornithology. The funders had no role in study design, data collection and analysis, decision to publish, or preparation of the manuscript.

==============================
The birds-of-paradise (Aves: Paradisaeidae) are a quintessential example of elaborate ornamental diversification among animals. Ornamental evolution in the birds-of-paradise is exemplified by the presence of a highly integrated courtship phenotype, which is the whole package of plumage ornaments, behaviors and sounds that each species uses during courtship. Characterizing a species’ courtship phenotype is therefore a key part of evolutionary and taxonomic investigation in the group. With its unprecedented transmogrification from bird-like form into something abstract and otherworldly, the courtship phenotype of the Superb Bird-of-Paradise, Lophorina superba, is one of the most remarkable of all. Recent research by Irestedt et al. (2017) suggests that the genus Lophorina is not a single species but is likely a complex of three allopatric species spanning the island of New Guinea: L. niedda in the Bird’s Head Peninsula of the west, L. superba throughout the central cordillera and L. minor in the Papuan Peninsula of the east. Of these, niedda is the most phenotypically divergent with plumage traits hypothesized to possibly produce differences in ornamental appearance during display. However, the whole courtship phenotype of niedda has not been documented and so the actual extent of differences in ornamental appearance during courtship remain unknown. Here we analyze the first audiovisual recordings of niedda and compare its courtship phenotype with superba to test the hypothesis of potential differences in ornamental appearance. Our main goals are to: (1) provide the first description of the courtship phenotype of niedda in the wild, (2) determine if and how the niedda courtship phenotype differs from superba and (3) evaluate any uncovered differences in light of niedda’s newly recognized species status. Our secondary goal is to provide a more thorough characterization of courtship phenotype diversity within the genus Lophorina to facilitate future comparative study within the genus and family. Results show that the niedda courtship phenotype differs substantially from superba in numerous aspects of ornamental appearance, display behavior and sound. We highlight six key differences and conclude that the new species status of niedda is corroborated by the distinctly differentiated ornamental features documented here. With full species status, niedda becomes the fourth endemic bird-of-paradise to the Bird’s Head region of Indonesian New Guinea (i.e., the Vogelkop Peninsula), a fact that underscores the importance of this region as a center of endemic biodiversity worthy of enhanced conservation protection.

Introduction

The birds-of-paradise (Aves: Paradisaeidae) have evolved some of the most elaborate diversification in ornamental mating displays among animals (Frith & Beehler, 1998; Laman & Scholes, 2012). A key innovation behind ornamental diversification in the birds-of-paradise is the evolution of a functionally integrated courtship phenotype (Scholes, 2008a). A species’ courtship phenotype is the whole package of plumage ornaments, behaviors and sounds used during courtship. In males, it encompasses the portion of the phenotype that is evaluated by females during courtship and in females it is the portion devoted to mate evaluation and selection (Bostwick, Harvey & Scholes, 2017). Characterizing a species’ courtship phenotype is an important part of understanding evolutionary patterns and taxonomic assessment in the birds-of-paradise (Scholes, 2008b; Scholes, Beehler & Laman, 2017; Scholes, Gillis & Laman, 2017).

Among the approximately 40 species of birds-of-paradise, one of the most remarkable male courtship phenotypes is that of the Superb Bird-of-Paradise, Lophorina superba (Frith & Frith, 1988; Laman & Scholes, 2012). Most of what is known about courtship in L. superba comes from older descriptions of captive birds (Manson-Bahr, 1935; Morrison-Scott, 1936; Seth-Smith, 1936; Timmis, 1968) and two accounts of wild birds from the Adelbert Range (Frith & Frith, 1988) and the central cordillera of western Papua New Guinea (Whiteside & Feignan, 1998). From these sources and film footage of L. superba from near Tari, Papua New Guinea obtained for the 1996 documentary film Attenborough in Paradise, Frith and Beehler synthesized all knowledge of courtship and mating behavior for L. superba (Frith & Beehler, 1998). The picture that emerged revealed two basic types of courtship display typically given from a horizontal free fall in the forest understory: a suite of relatively simple behaviors called initial display activity and the iconic transmogrifying display, called the high intensity display, which features a highly specialized ornamental nape cape that is spread laterally to form a super black (McCoy et al., 2018) oval-shaped ruff that surrounds the entire head and upper body (Frith & Beehler, 1998). This otherworldly display has been likened in appearance to a black and blue bouncing “smiley” face-like form (Laman & Scholes, 2012).

Historically, the genus Lophorina has been treated as a single polytypic species with populations extending throughout mid-montane New Guinea from the Bird’s Head (Vogelkop) Peninsula in the west, through all of the central cordillera and to the Papuan Peninsula in the east with isolated populations in the Adelbert Range and Huon Peninsula in the northeast (Frith & Beehler, 1998; Frith & Frith, 2009). The Bird’s Head and Papuan Peninsula populations exhibit geographic variation in female head color (Cracraft, 1992; Frith & Beehler, 1998; Pratt & Beehler, 2015) and the Bird’s Head population has minor variation in male ornamental plumage—i.e., no black spots on the feathers of the breast shield ornament (Cracraft, 1992; Pratt & Beehler, 2015). Recently, the Bird’s Head population has also been identified as being vocally distinct from the others, but comparative acoustic details have not yet been examined (Beehler & Pratt, 2016).

Recent analysis of DNA sequence data and external morphology from museum specimens suggest that the genus Lophorina is likely comprised of three species (Irestedt et al., 2017). The newly delimited species are: L. niedda (with subspecies niedda and inopinata) in the Bird’s Head (Vogelkop) Peninsula, L. superba (with subspecies: superba, addenda and latipennis) throughout the central cordillera, Adelbert Range and Huon Peninsula, and L. minor in the Papuan Peninsula (Irestedt et al., 2017). Of these, minor is the oldest and most genetically divergent lineage, but niedda is the most phenotypically differentiated (Irestedt et al., 2017). For example, the breast shield feathers of male niedda are not as laterally extended as they are in superba and minor and the outer ornamental cape feathers of niedda are more elongated with outwardly-curved and narrowly rounded tips with reduced hair-like fringing (Irestedt et al., 2017). As suggested by (Irestedt et al., 2017), the plumage differences of niedda might be expected to produce significant differences in ornamental appearance during display and thereby further validate its newly revised species status, but the courtship phenotype of niedda has not been documented in the wild.

Here we analyze the first audiovisual recordings of courtship behavior in niedda and compare its courtship phenotype with superba based on analysis of audiovisual recordings from the Macaulay Library at the Cornell Lab of Ornithology and several documentary films. Our aim is threefold: (1) to provide the first description of the courtship phenotype of niedda in the wild, (2) test for potential differences in ornamental appearance by determining if and how the niedda courtship phenotype differs from superba and (3) evaluate any uncovered differences in light of niedda’s recently recognized species status. An additional goal is to provide a more thorough characterization of courtship phenotype diversity within the genus Lophorina to facilitate future comparative study within the genus and among species in the family.

Materials & Methods

Field observations and audiovisual recordings of two adult male L. niedda were made near the village of Mingre in the Arfak Mountains of West Papua, Indonesia during June-November 2016 and 2017. Multiple simultaneous recordings were made with remotely and manually operated video cameras and audio recorders from within observation blinds. Video was recorded with RED Epic-W Helium and Canon 1DX Mark II cameras fitted with a range of Canon lenses. Audio was recorded using a Marantz PMD 661 and Sennheiser microphones. Focal length, frame size and frame rates of video recordings varied according to the particular camera-lens combination used and particular environmental conditions, but ranged between 50–800 mm, 4 k–8 k and 24–240 p respectively.

For the video analysis, we screened over 200 raw field recordings using the REDCINE-X Professional and Canon EOS MOVIE Utility software applications. Clips with display behaviors were identified and analyzed frame-by-frame to identify the components of courtship described in the results. Recordings containing practice displays and displays given to females were used for the detailed analysis (Table S1). In total we analyzed 30 practice displays and 14 displays to females. Voucher recordings were archived in the biodiversity media collections of the Macaulay Library at the Cornell Lab of Ornithology (http://www.macaulaylibrary.org) as accession number ACC4396. Vouchers are cited in the Results using Macaulay Library catalog numbers and can be viewed and/or requested online by searching the Macaulay Library database for catalog numbers ML487530 to ML487569 (see also Table S1). Media can also accessed directly using the following URL: https://macaulaylibrary.org/asset/xxxxxx where “x’s” are replaced with the desired catalog number.

Video frames highlighting important components of the courtship phenotype were extracted as tiff files for creating figures. Modifications to figure images included minor adjustments to improve clarity (e.g., cropping, and small adjustments in levels, brightness and contrast).

For the acoustic analysis, archival quality audio files from the Macaulay Library (44 khz, 16-bit .wav; Table S1) were analyzed with the sound analysis software package Raven Pro version 1.5 (Cornell Lab of Ornithology, Ithaca, NY, USA).

Video used for the analysis of L. superba came from existing archival video and sound recordings from the Macaulay Library (Table S1) of two individual adult male superba superba (ML487529, ML487528 and ML458167) and one individual adult male superba latipennis (ML458001 –ML458003). All of the recordings with data used in the analysis are cited in the Results and can be viewed and/or requested online via the Macaulay Library database as described above.

To broaden our review of potential interspecific variation within superba, we also reviewed broadcast video footage obtained by film crews during the production of three documentary films: (1) Planet Earth (2006) produced and distributed by the BBC, (2) Nature: Birds of the Gods (2011) distributed by PBS, and (3) Designed to Dance: Birds of Paradise (2011) produced by NHK Wildlife. Through communications with the producers and/or with local informants at the filming locations, we know that all three the documentary sequences were filmed near the town of Tari in Hela Province, Papua New Guinea and therefore all represent the subspecies L. superba addenda. Clips containing the segments we reviewed and/or full episodes for each film can be viewed online through a variety of sources (e.g., the Planet Earth segment is viewable on the YouTube channel of BBC Earth: https://youtu.be/nWfyw51DQfU?t=2m10s and Birds of the Gods can be viewed on the Nature website: http://www.pbs.org/wnet/nature/bird-of-the-gods-full-episode/13378/?button=fullepisode from 40:41 to 42:58). Due to copyright restrictions, we could not include comparable images of addenda in our figures, but unless otherwise noted, the courtship phenotype of addenda was found to be identical to those of superba and latipennis documented here.

Results

Initial display activity

Our analysis identified three distinct types of initial display activity in niedda, which broadly correspond with those of superba. In the order in which they occur within the courtship sequence, they are: advertisement, horizontal and pointing displays. Each display is described in detail below. Our analysis did not identify substantial variation in the behaviors, postures and ornaments within the superba complex (i.e., among s. superba, s. addenda and s. latipennis) and so we use latipennis as the frame of reference to describe and illustrate the superba complex (but see Fig. S1 for details of superba superba). Although the three types of initial display activity broadly correspond with those of superba, our analysis found that each display of niedda has numerous distinctive elements.

Advertisement display

The central features of this display involve the male excitedly looking up and around from side to side (presumably for a potential mate) while repeatedly flicking his cape forward in front of his head and calling (video vouchers: superba ML458000 from 0:44 to end; niedda ML487533). Upon arrival to the fallen log used for display, the male stands in an alert upright posture with head up, bill forward and tail down (Figs. 1A & 1B and S1A). The iridescent crown feathers are pushed (anteriorly) toward the bill so that they do not visibly reflect any blue and the entire head appears black from all directions. In this configuration, the ornamental eye-spots are not visible (Figs. 1A & 1B and S1A & S1B). In niedda, the male typically arrives and departs from his display log by flying low through the forest understory rather than by dropping down from or flying up to the forest canopy immediately above. It is unclear if superba does the same with the available data.

Figure 1 Advertisement display of superba and niedda.

The central features of this display involve the male excitedly looking up and around from side to side while calling and flicking the cape in front of the head. In superba (A), the cape is held folded over the back and wings, without being spread, such that the terminal tips of the cape reach to, or extend beyond, the base of the tail (see arrow). The feathers of the breast shield are held close to the breast so that the color of ornament is not visible. In neidda (B), the cape is spread widely over the upper back such that the terminal tips extend conspicuously to the sides (see arrow) leaving the wing tips and base of the tail uncovered. The breast shield is not held as tightly against the breast and its shorter and narrower lateral tips do not fold back toward the torso as in superba. The terminal tips of the neidda breast shield curve gently outward, which makes the surface of the breast shield slightly concave. The primary movement is a frequent forward “flicking” of the cape, which results from turning the head down and to the side into a partially open wing in order to perform a rapid false wing preen motion. In superba (C), the cape largely retains its compact form when flicked forward, whereas in niedda (D), the cape fans out distinctively over the head. Image credit/source: (A and C) Edwin Scholes/ML458003 and (B and D) Tim Laman/ML487533.

In superba, the main posture of the advertisement display has the nape cape folded over the back and wings, without being spread, such that the terminal tips of the cape reach to, or extend beyond, the base of the tail completely covering the wings (Fig. 1A). In neidda, the configuration differs in that the cape sits spread over the upper back such that the terminal tips extend conspicuously to the sides leaving the wing tips and base of the tail uncovered (Fig. 1B).

In superba, the feathers of the breast shield are held close to the breast and upper body so that, for the most part, the color of the ornament is not visible, especially as viewed from above (Figs. 1A and S1A). One exception is during high-intensity calling in which case the breast is thrust forward causing the breast shield to be pushed outward and made visible (Fig. S1B). In contrast, the breast shield of neidda is not held as tightly against the breast and its shorter and narrower lateral tips do not fold back toward the torso as in superba (Fig. 1B). The terminal tips of the neidda breast shield curve gently outward, which makes the surface of the breast shield slightly concave (Fig. 1B). In both superba and niedda, the bright yellow color of the gape becomes visible while calling (Figs. 1A & 1B and S1B).

Beyond repeated side to side head turning, the primary movements of the display are a frequent forward “flicking” of the cape, which results from turning the head down and to the side and into a partially open wing in order to perform a rapid false wing preen motion (Figs. 1C and 1D). The extremely fast movement of the false-preen causes the cape to be quickly flicked forward over the downward turned head in a highly exaggerated and ritualized fashion. In superba, the cape largely retains its compact form when flicked forward whereas in niedda, the flicked cape is fanned out over the head (Figs. 1C and 1D).

Horizontal display

This display is frequently interspersed throughout bouts of the advertisement display and involves the diagnostic adoption of a rigid horizontal posture with the neck outstretched, head lowered, and bill pointing forward to be in line with the body axis (video vouchers: superba ML458000 (from 0:33–0:44) and ML458001 (from 0:27–0:34); niedda ML487558 from 0:00–0:30). The tail is also lifted so that the whole body forms a horizontal line from tip of the bill to tip of the tail (Figs. 2 and S1C).

Figure 2 Horizontal display of superba and niedda.

This display involves the diagnostic adoption of a low rigid horizontal posture with the neck outstretched, head lowered, bill pointing forward and tail lifted to be horizontal so that the whole body forms a horizontal line from tip of the bill to tip of the tail. Crown feathers are pushed forward so that the head remains all black in appearance and the ornamental eye-spots are not visible. In superba (A), the cape is folded over the back to the base of the tail and is not spread. Arrow “C” points to the terminal tips of the cape feathers. The longest feathers in the breast shield of superba stick out conspicuously to the sides while the central breast feathers are sleeked against the body so that no color is visible. The dark outwardly pointed tips give the appearance of being small pointy “wings” sticking out from the sides of the breast. Arrow “BS” points to the terminal tips of the breast shield feathers on the bird’s right side. In niedda (B), the cape is spread widely over the upper back to form a slightly raised narrow crescent with decurved “wings” spread to each side and with tips that droop to the log. Arrow “C” points to the terminal tips of the cape feathers on the bird’s right side. The breast shield of niedda is sleeked against the body with no color visible. Unlike the arrangement in superba, the longest breast shield feathers of niedda are not as noticeable because of how they are obscured by the wing-like position of the cape. Image credit/source: (A) Edwin Scholes/ML458003 and (B) Tim Laman/ML487563.

In most instances, the crown feathers are pushed forward so that the head remains all black in appearance and the ornamental eye-spots are not visible (Fig. 2). However, the eye-spots are occasionally revealed during practice displays or if the bird is about to transition into the pointing display (see below), but then concealed again as the horizontal display continues or the bird ruffles its plumage and returns to a bout of advertisement display.

In superba, the cape is folded over the back to the base of the tail and is not spread (Figs. 2A and S1C). In a position that creates a very different appearance, the cape of niedda is held over the back, but spread so that it forms two decurved “wings” that emanate from each side and with tips that droop to the log (Fig. 2B). At times, the cape is partially lifted over the back so that it forms a slight arch behind the head.

In superba, the longest (lateral) feathers of the breast shield stick out conspicuously to the sides and the central most breast feathers are sleeked against the body so that no color is visible, especially when viewed from above (Fig. 2). Because the body is in elongated horizontal posture, only the blackish underside (ventral side) of the breast feathers project outward. The dark outwardly pointed tips give the appearance of being small pointy “wings” sticking out from the sides of the breast. In niedda, the breast shield is also sleeked against the body with no color visible (Fig. 2B), however unlike the arrangement in superba, the longest breast shield feathers do not look like “wings” because the wing-like position of the cape obscures them (Fig. 2B).

The main movement of the horizontal display is ritualized hopping “run” along the log to change position and orientation (e.g., see ML487557 from 1:16 to end and ML487563 from 0:00–0:25). The hops use both feet at the same time and are typically initiated with a quick wing-flick just prior to the first hop. Although less frequent than in advertisement display, the false wing-preens that cause frontal cape flicking are also given.

Pointing display

The final display in the initial display activity sequence begins from the posture of the horizontal display, which is the most common, or following an advertisement display vocalization. As the name suggests, the primary feature of this display involves the adoption of a ritualized pointing posture where the bill points skyward as the male’s attention is oriented toward a potential, or actual, female (video vouchers: superba ML458002 from 0:22–0:53 and ML458003 from 0:50–0:56; niedda ML487538 from 0:00–0:03 and ML487540 from 2:10–2:23). In superba, the body is held in an upright posture, with the upper body lifted from the horizontal to be erect as if the bird is “standing at attention” (Fig. 3A). In niedda, body posture is more horizontal with the upper body closer to the log and the head tilted back so that the bill points upward (Figs. 3B and 4).

Figure 3 Pointing display of superba and niedda.

This display involves a ritualized pointing posture where the bill points skyward. The feathers of the fore crown are pushed back against forehead the while the hind crown feathers remain pushed forward, which makes the reflective fore crown visible. The narial tufts are lifted outward from both sides of the bill, which creates the distinctive eye-spot ornament when aligned with the fore crown feathers and viewed head-on. In superba (A), the body is held upright, as if the bird is “standing at attention” with breast shield prominently displayed. Narial tufts are wedge-shaped and do not extend beyond the width of the head. The cape remains folded over the back. In niedda (B), the body posture is more horizontal with the head tilted back so that the bill points upward. Narial tufts are elongate with tips that extend out beyond the width of the head. The cape is spread over the back to form the conspicuous decurved wing-like protrusions, which obscure the breast shield. Movements include a ritualized flapping of the cape over the back. In superba (C), the cape-flap occurs when the wings are partially lifted flicked open over the back in coordination with a quick foot stomp, which causes the cape to flap up-and-down such that the terminal tips of the cape feathers rise above the height the of the head. In niedda (D), there is a foot stomp, but wings are only raised slightly up-and-down over the back, which causes the terminal tips of the cape feathers stay at or below the level of the head. Image credit/source: (A and C) Edwin Scholes/ML458003 and (B and D) Tim Laman/ML487538.

A key feature of this display is how the feathers of the fore crown are pushed back (posteriorly) to be flattened against forehead the while the hind crown feathers remain pushed forward (anteriorly). This result of this repositioning is that the highly reflective fore crown feathers become visible, changing the appearance from all-black to an iridescent electric blue (Figs. 3 and 4). At the same time, the feathers of the narial tufts are also pushed outward from both sides of the bill, which when aligned with the blue fore crown feathers, and viewed head-on, create the highly distinctive eye-spot ornament. When visible, the eye-spots reflect so brightly relative to the super-black plumage surrounding them that they look somewhat like a pair of headlights (Figs. 3 and 4). It should also be noted that the shape of the erected narial tufts differs between niedda and superba. In superba, the tufts are shorter and thicker, almost wedge-shaped, and do not extend beyond the width of the head (Fig. 3A). In niedda, the tufts are narrow and elongate with tips that extend out beyond the width of the head (Fig. 3B).

Figure 4 Pointing display of niedda as viewed by a female.

The white arrow points to a female plumaged bird (and presumed female) observing the male on his display log. Note how the breast shield remains sleeked against the breast and is effectively invisible—its lateral tips concealed (shadowed) by the protruding “wings” of the cape. In this context, the eye-spots reflect so brightly relative to the super-black plumage of the rest of the male that they look like a pair of headlights “shining” toward the female. This “headlight pose” with the wing-like cape results in a very different appearance from the pointing display of superba. Image credit/source: Tim Laman/ML487540.

As with the previous behaviors, the arrangement of the cape differs substantially between superba and niedda. In superba, the cape remains tightly folded over the back and is not noticeably spread (Fig. 3A). In niedda, the cape is always prominently spread broadly over the back to form the conspicuous decurved wing-like lateral protrusions that are a hallmark of the niedda courtship phenotype (Fig. 3B).

During the pointing display, the breast shield of superba, though not as outstretched as in the cape presentation display (see below), is nevertheless quite prominent with its reflective blue color visible (Figs. 3A and S1D). In niedda, the breast shield remains sleeked against the breast and largely invisible—its lateral tips concealed (shadowed) by the protruding “wings” of the cape—when viewed from the vantage point of an approaching female (Fig. 4).

The main movements during the pointing display involve a ritualized flapping of the cape up-and-down over the back and occasional reorientation along the log to “track” the position of a female as she moves around nearby or approaches. The details of the cape-flap differ between superba and niedda. In superba the cape-flap occurs when the wings are partially lifted and flicked open over the back in coordination with a quick pronounced foot stomp, which causes the cape to flap up-and-down over the back in an exaggerated way such that the terminal tips of the cape feathers rise above the height the of the head (Fig. 3C). In niedda, there is a foot stomp, but wings are not flicked open, and are only just raised slightly up-and-down over the back, which results in the cape flapping up, but not as high, and such that the terminal tips of the cape feathers stay at or below the level of the head (Fig. 4D).

If the pointing display does not transition into a bout of high intensity display (see below), the fore crown feathers may be pushed forward and back, which causes the eye-spots to disappear and reappear or to “flash” on and off. In that scenario, the breast shield of superba is sometimes sleeked against the body again, which cause the reflective blue color to visually “blink off” as well. The cape-flapping may also revert into the pseudo wing-preen cape flicks that are a common feature of the advertisement display.

A noteworthy behavior associated with the initial display activity of niedda is a leaf tossing behavior (e.g., ML487565 from 1:24–1:35). For this behavior, the male picks up a leaf from the surface of the display log and tosses it into the air in such a way that it often lands on the log nearby. It is unclear if this behavior is a rudimentary form of court clearing behavior similar to that known from Parotia species or if it is some form of ritualized visual display like those of Parotia carolae, which uses leaves as props during certain displays (Scholes, 2006).

High intensity display

Our analysis identified one high intensity display in both species, which we call the cape presentation display. Our analysis did not uncover substantial variation in the behaviors, postures and ornament forms among s. superba, s. addenda and s. latipennis, but did find minor variation in cadence of bouncing and snapping sounds (see below). As with the initial display activities, we use s. latipennis as the standard frame of reference for the superba complex and details on s. superba are provided in Fig. S1. As with the initial display activities above, our analysis of niedda found it to have numerous distinctive elements, which are described below.

Cape presentation display

The visual form and movements of this display comprise the quintessential features of courtship in the genus Lophorina (video vouchers: superba ML458003; niedda ML487557 and ML487562). When performed to a female, the cape presentation display is typically initiated from the posture of the pointing display—i.e., Fig. 3A superba and Figs. 3B and 4 niedda). However, if a female appears suddenly without stopping on a nearby perch on her approach, and during practice displays (no female present), the male will often transition directly into the cape presentation from any of initial display activities (e.g., in ML487557 the pointing display is omitted and the male appears to be caught off guard by the sudden arrival of the female).

The cape presentation display begins when the male perceives a female making her approach toward the log. In both superba and niedda, the cape-flapping of the pointing display (see above) increases in intensity. In niedda, the increase is greater as wings are lifted higher over the back and then quickly flicked open and shut in the exaggerated manner of superba during pointing (Figs. 5A and 5B). As in the pointing of superba, the cape feathers of niedda also flap higher such that the tips exceed the height the of the upward pointing bill (Fig. 5B). In conjunction with the exaggerated wing-flick/cape-flaps, the head is tilted back more sharply, the bill is also opened widely to reveal the bright yellow gape and the breast is thrust forward so that the feathers of breast shield ornament become fully expanded and conspicuous—its reflective surface angled upward toward the approaching female (Figs. 5A and 5B). The point at which the male presents his bright yellow gape is the best landmark for identifying of the start of the cape presentation display even though there is often a bout of the exaggerated wing-flick/cape-flaps preceding it.

Figure 5 Cape presentation display of superba and niedda.

The visual form of this display is the quintessential feature of courtship in the genus Lophorina. In both superba and niedda (A and B), cape-flapping increases in intensity. In niedda (B), the increase is greater because the wings are lifted higher over the back and then flicked open and shut in the exaggerated manner of superba. Simultaneously, the breast is thrust forward so that the breast shield becomes fully expanded and conspicuous with its reflective surface angled toward the approaching female. In superba (C), the cape is lifted into presentation position before to female arrives on the log. In L. niedda (D), wing-flick/cape-flaps continue for several more bouts after the female arrives and before the cape is put into presentation position. The appearance of the cape presentation phenotype differs dramatically between superba (E) and niedda (F). Image credit/source: (A, C and E) Edwin Scholes/ML458003, (B) Tim Laman/ML487538 (D and F) Tim Laman/ML487557.

The visual effect of the exaggerated wing-flick/cape-flaps with the breast shield fully revealed is more pronounced in niedda than superba because of the stark difference in the appearance of the breast shield during the pointing display (e.g., compare breast shield positions in Figs. 3A and 3B to those in Figs. 5A and 5B). In both species, the male will reorient himself as needed to track the position of the approaching female and keep his eye-spots and breast shield directed toward the female (e.g., video ML487530).

In niedda, bouts of wing-flick/cape-flapping coincide with a distinctive very rapid side-to-side wobbling of the head. As viewed from the front (i.e., the POV of the approaching female) the head wobbling has the effect of making the eye-spots flicker or shimmer. It is possible that the wobbling head movement may also be present in superba, but is not discernible from the available data.

Figure 6 Detailed comparison of the cape presentation form of superba and niedda.

In superba (A), the shape of the cape presentation form is oval—i.e., wider than it is high. The edge of the opened cape has a smooth fuzzy appearance, which is created by the distinct hair-like fringing along the terminal tips of the individual feathers that comprise the cape. In niedda (B), the shape of the cape presentation form is distinctly crescent shaped—i.e., it has a curved sickle shape that is broad in the center and tapers to a point at each end. The contour of the crescent is slightly ribbed and lacks the fuzzy appearance of superba. The shape and appearance of the breast shield also differs between the cape presentation phenotype of superba and neidda. In superba (A), the breast shield is wider with thicker terminal ends that make a relatively flat surface with a fairly straight edge along the top of the ornament. In neidda (B), the shorter and narrower terminal tips of the breast shield curve outward so that the surface of the ornament is slightly concave and the top edge is angled downward at the sides. This difference results in niedda looking like somewhat like “frowning face” whereas superba has more of a “smiling face” look. Also, the thinner profile of the narial tufts of niedda often makes the eye-spots look like they have thin blue “eyebrows” just above each eye-spots. Because the narial tufts fo superba are shorter and do not extend over the top margin of the eye-spot, “eyebrows” are possible. Image credit/source: (A) Edwin Scholes/ML458003 and (B) Tim Laman/ML487563.

Next comes the transition into the main part of cape presentation display—i.e., the presentation of the fully opened cape itself. In superba, the cape is lifted into its presentation position well before to female arrives on the log (Fig. 5C). In L. niedda, the wing-flick/cape-flaps with head wobbles continue for several more bouts after the female has arrived on the log and approaches the male become face-to-face with him (Fig. 5D). As the neidda female moves closer, the male will back up slightly and crouch lower before deploying his cape for full presentation (e.g., videos ML487530, ML487538, and ML487562).

Although the timing of cape deployment differs between superba and niedda, the cape of both is lifted into the full display position in largely the same way. First, it is spread wide, fan-like, over the back and then it is raised (anteriorly) and pushed over the head until it creates the super-black (McCoy et al., 2018), slightly concave, cape presentation display form that surrounds the entire head and upper body (Figs. 5E & 5F and S1E).

Once in position however, the form and appearance of the cape presentation phenotype differs dramatically between superba and niedda. In superba, the shape of the cape presentation form is oval—i.e., wider than it is high (Figs. 6A & S1F). In comparison, the shape of the cape presentation form of niedda is distinctly crescent shaped—i.e., it has a curved sickle shape that is broad in the center and tapers to a point at each end (Fig. 6B). In superba, the border edge of the fully opened cape has a smooth fuzzy appearance (Fig. 6A), which is created by the distinct hair-like fringing along the terminal tips of the individual feathers that comprise the cape. The border edge of the crescent in niedda is slightly ribbed and lacks the fuzzy appearance of superba (Fig. 6B). Correspondingly, the terminal tips of the niedda cape feathers have very minor hair-like fringing. The iconic oval shape of superba exhibits very little variation within the superba complex (e.g., Fig. 6A and Fig. S1F for a comparison of L. s. latipennis and L. s. superba and refer to the documentary films described in the Methods for comparative imagery of L. s. addenda).

The shape and appearance of the breast shield also differs between the cape presentation phenotype of superba and neidda. In superba, the breast shield is wider with thicker terminal ends that make a relatively flat surface with a fairly straight edge along the top of the ornament (Fig. 6A). In neidda, the shorter and narrower terminal tips of the breast shield curve outward so that the surface of the ornament is slightly concave and the top edge is angled downward at the sides (Fig. 6B). This difference results in niedda looking like somewhat like “frowning face” whereas superba has a “smiling face” look. The “smiling” vs “frowning” face appearance is further enhanced by the difference in how the eye-spot ornaments are created. In superba, the the narial tufts are thick and relatively short whereas in niedda they are narrow and longer (i.e., extend beyond the width of the head). This difference has the effect of making the eye-spots of niedda appear smaller with a straighter line along the top edge, which gives the “eyes” the appearance of having slightly droopy “eyelids” compared with the always larger and circular appearance of the “eyes” in superba (Figs. 6A and 6B). In addition, when alignment of the feathers is not perfect relative to the viewer, the thinner profile of the narial tufts often makes the eye-spots of niedda look like they have a thin blue “eyebrow” just above each eye-spots (Fig. partially visible in Fig. 5F). Because the narial tufts fo superba are shorter and do not extend over the top margin of the eye-spot, “eyebrows” are never visible nor even possible.

Throughout the cape presentation display, the transformed oval-shaped cape presentation form of superba is repeatedly lifted up and down, or bounced, with outstretched and bent legs (e.g., ML458003). The repeated up and down movements coincide with a series of alternating shuffle–hop steps to the side, which create the distinct bouncing cadence: up–down shuffle–hop, first to one side and then the other. When a female is present, the male’s bouncing shuffle–hop movements revolve around the female in either a semi-circular or full circle path.

In niedda, the overall appearance of the rotational movement during the cape presentation display is very different from that of superba. Instead of the characteristic bouncing of superba, the feet of niedda move in a faster, more pronounced, side-step motion without bouncing and with no visible up and down movement of the body. This creates the effect of niedda appearing to move in a quick but steady glide, rather than bounce, around the female in semi- or full circle (e.g., ML487557 and ML487562).

Figure 7 Wing-flick snaps of superba and niedda.

In superba (A), distinct snapping sounds are a characteristic acoustic component of the cape presentation display and occur rhythmically with the up–down bouncing movement on the down beat of the bounces, but not on every bounce. In superba latipennis, the cadence of snaps and bouncing is down-snap-up, down-snap-up, down-up such that snaps occur in doublets with a distinct pause in between, which sounds like “snap-snap (pause) snap-snap”. In niedda (B), where the rotational movement lacks the bouncing of superba, the overall pattern and effect of the wing/tail flicks and snaps is very different because they occur at a regular interval that is not timed with any particular aspect of the rotational movement. The snaps of niedda are given at uniform beat of approximately one snap per second for a steady “snap-snap-snap” sound. Audio credit/source: (A) Edwin Scholes/ML458003 and (B) Edwin Scholes/ML8576351.

Along with the bouncing (superba) or smooth quick (niedda) rotational movement of the cape presentation form along the display log, the tail of both species is lifted to about 45 degrees above the plane of the back and rectrices are fanned. The secondary feathers of the wings are spread such that they become partially intertwined with the rectrices (see Fig. S1E). In one explosive motion, the wings are flicked open and shut to the sides, which causes a violent interaction among the secondaries and rectrices. In superba, the explosive wing-flicks occur rhythmically with the up–down bouncing movement and coincides with the distinct snapping sounds that are a characteristic acoustic component of the display. In superba, the wing-flicks happen on the “down beat” of the bounces, but not on every bounce. In s. latipennis, the cadence of snaps and bouncing is down-snap-up, down-snap-up, down-up such that snaps occur in doublets with a distinct pause in between, which sounds like “snap-snap (pause) snap-snap” (Fig. 7A). In niedda, where the rotational movement lacks the bouncing of superba, the overall pattern and effect of the wing/tail flicks and snaps is very different because they occur at a regular interval that is not timed with any particular aspect of the rotational movement. The snaps of niedda are given at uniform beat of approximately one snap per second for a steady “snap-snap-snap” sound (Fig. 8B).

In s. superba, the cadence of the snapping is more frequent with less obvious choreography among the snaps and bouncing movements. In s. addenda, to the extent discernible from the documentary video sources, both the cadence of the bouncing and snapping seems to be more frequent with wing-flicks (and presumably the snaps) occurring with each downward bouncing stroke, “down-snap-up, down-snap-up, down-snap-up”. In both niedda and superba, the exact relationship between the wing movement and snapping sound production is unclear and warrants further investigation using high-speed videography.

In both superba and niedda, the cape presentation display ends when the female moves quickly away from the male such that he is unable to continue rotating around her, or when the female solicits copulation. At times the female moves along the log ahead of the male and at other times she will fly away from the log entirely. When females depart, males of both superba and niedda maintain the full cape presentation posture directed toward her and slowly lower it down over the back to readopt the pointing posture. In niedda, as the female departs, the male opens his bill to reveal the bright yellow gape once again. This end of display reveal of the gape was not observed in superba.

Figure 8 Advertisement vocalizations of superba and niedda.

Advertisement vocalizations of both niedda and superba are phrases of 2–8 notes, with five notes common. While the number of notes is similar in both species, superba (A) gives a characteristic harsh raspy screech. The acoustic quality of niedda (B) vocalizations are very different. The basic note of niedda is a high-pitched nasal “yiap” sound, which has a piercing whistle-like quality that ascends and descends slightly in frequency. Audio credit/source: (A) Edwin Scholes/ML458003 and (B) Edwin Scholes/ML85763571.

If a female solicits copulation (e.g., ML487568 from 0:07–0:17), she beings fluttering her wings as the male rotates around her and then will tip forward with breast low and tail high. At this point, the male circles closely and mounts the female while maintaining the fully open cape posture. The cape is held in the open position during copulation and wing-snaps are given as part of, or just after, the wing flapping that occurs with copulation. After copulation, the male dismounts from the back of the female and resumes the cape presentation display directed at the female (e.g., ML487562). If she departs the log, the male holds the opened cape posture for a few seconds and the slowly lowers it and often returns to the pointing display and sometimes even reverts backwards through the entire sequence from pointing, to horizontal to advertisement displays.

Vocalizations

Advertisement vocalizations of both niedda and superba are phrases of 2–8 notes, with five notes being the most typical (video vouchers: superba ML458003 and niedda ML487530; audio voucher: niedda ML85763571). While the number of notes is similar in both species, the acoustic quality of niedda is very different from the characteristic harsh screeching notes of superba (Fig. 7). The basic note structure of superba is a harsh raspy screech with little acoustic structure (Fig. 7A). In comparison, the basic note of niedda is a high-pitched nasal “yiap” sound, which has a piercing whistle-like quality that ascends and descends slightly in frequency (Fig. 7B). Note that variation in niedda includes a version with a more constant pitch (i.e., the frequency does not ascend and/or descend), which sounds more like “yeep” instead of the up-and-down frequency of the “yiap” notes. Another common variant is slightly higher-pitched and distinctly descending to sound more like “iap” rather than “yiap” (Fig. 7B). While all three of the common note types can be present in a single phrase, they are usually not given that way. Phrases with a string of 5–6 “yiap” notes are the most common.

Discussion

The primary purpose of this study was to provide the first description of the courtship phenotype of Lophorina niedda in the wild and determine if and how it differs from the courtship phenotype of L. superba with special consideration of what differences, if found, would mean in light of niedda’s newly recognized species status. A secondary goal was to provide a more thorough characterization of courtship phenotype diversity within the genus Lophorina in order to facilitate future in-depth comparative study.

Our results show that the niedda courtship phenotype differs from superba in numerous appearance, behavioral and acoustic components. Yet at the same time, the niedda courtship phenotype does not contain entirely unique courtship behaviors relative to superba—i.e., both have the same higher-order courtship phenotype architecture made up of three relatively simple initial display activities (advertisement, horizontal and pointing displays) and one complex high intensity display (cape presentation). In short, both superba and niedda share the same underlying courtship phenotype substructure, but differ in many details.

The six most diagnostic and noteworthy differences in the courtship phenotype of niedda compared with superba are: (1) the protruding wing-like position of the ornamental cape during the horizontal and pointing displays (Figs. 2B and 3B); (2) how the breast shield is obscured during the pointing display, which visually emphasizes the “headlight-like” appearance of the eye-spot ornament as viewed by a female observer (Figs. 3 and 4); (3) how during the cape presentation display, the cape is not fully opened until after the female approaches at close range on the display log (Fig. 5D); (4) the striking difference in appearance of the fully opened cape presentation form—i.e., the distinctive crescent shape, the ribbed contour, the “frowning face” look of the breast shield and the way in which the eye-spots sometimes have “eyebrows” (Fig. 6); (5) how during the during of the cape presentation dance, the male revolves around the female with a smooth quick gliding motion (instead of the characteristic bouncing of superba) and with a constant (rather than doublet) wing-snap beat (Fig. 7); and (6) a piercing whistle-like vocalization instead of the characteristic raspy screech of superba (Fig. 8).

While the differences uncovered here are compelling, there are nevertheless several caveats to be considered. For example, our results are based on relatively low individual sample sizes for both niedda and superba and therefore may not reflect the full scope of intraspecific variation. However, with the basic framework now in place, future studies will be able to assess this more critically as larger sample sizes are obtained. It should also be noted that while we did not find substantial differences within superba (i.e., among L. s. superba, L. s. addenda and L. s. latipennis), sample sizes were limited, based on incomplete primary sources (for L. s. addenda) and included an outlying population from the Adelbert Mountains (L. s. latipennis) as the main frame of reference for all superba. To this last point, given the relatively small and geographically isolated population in the Adelbert Mountains, it is possible that further observations and larger sample sizes could reveal diagnostic differences between it (Adelbert) and cordillera populations. As noted in the Results, there are indications of differences in cadence of wing snaps and bouncing tempo of the cape presentation display among L. s. superba, L. s. addenda and L. s. latipennis. Quantifying these differences requires additional field observation and larger sample sizes.

What do the differences uncovered here mean with regard to the species status of niedda? We think the distinctive courtship phenotype confirms the conclusions of Irestedt et al. (2017) regarding the species status of niedda. With the mounting genetic, external morphological and now full courtship phenotype evidence, there can now be little doubt that population of Lophorina inhabiting the Bird’s Head Peninsula of western New Guinea is a distinct evolutionary lineage that is both geographically and reproductively isolated from the nearest population of superba in the Kobowre (Weyland) Mountains of the western cordillera. Furthermore, because all the differences described here are directly related to mating preferences of females and have likely evolved through divergent aesthetic preferences of females, it seems probable that niedda and/or superba females would not find males of the other type as attractive as males of their own type. However, reciprocal exclusivity in female mating preferences is not required to meet the standards of species delineation in the birds-of-paradise. With the exception of the newly defined L. minor, for which there is not enough data to make a determination, bird-of-paradise species are defined by the degree of differences in their courtship phenotypes. In other words, courtship phenotype disparity is the key criterion for species diagnosis in the birds-of-paradise. By this yardstick, L. niedda is clearly a distinct species.

In addition to characterizing the courtship phenotype of niedda for the first time, our results also provide a more complete picture of the superba courtship phenotype. For the first time, we document that the initial display activity of superba is comprised of three distinct behaviors: the advertisement, horizontal and pointing displays (Figs. 1–3). Previous efforts to describe the displays of superba did not differentiate these components into distinct displays even though the basic patterns were described (Frith & Beehler, 1998). We also provide the most detailed (and vouchered) descriptions of the superba courtship phenotype to date.

The findings presented here, along with those of Irestedt et al. (2017) indicating that L. minor is the most genetically divergent lineage in the genus, highlight the need for more investigation of the courtship phenotype of minor. Are there differences in the courtship phenotype of minor that would shed light on its species status? Currently the only information about the courtship behavior of minor comes from incomplete observations of a single captive male that housed in the London Zoological Gardens during the late 1920’s and mid 1930’s (Manson-Bahr, 1935; Morrison-Scott, 1936; Seth-Smith, 1936). While lacking sufficient description to be fully comparable to the data presented here, it is nevertheless possible to glean several key details. For one, it is evident that minor does perform the cape presentation display. However, the presence/absence of initial display activities is unclear from the available data. In fact, when describing the cape presentation display, Manson-Bahr goes as far as stating, “there are no preliminary stages…the display is spontaneous and instantaneous” (Manson-Bahr, 1935, pg. 67). While undoubtedly possible, we feel fairly confident that some form of these initial behaviors were either overlooked or lacking by virtue of the conditions of captivity. Regarding the cape presentation display, it is clear from both written descriptions and illustrations that the form of the cape presentation is oval-shaped like superba, not crescent-shaped like niedda, with visible eye-spots made by the crown feathers (Morrison-Scott, 1936; Seth-Smith, 1936). The captive minor also performed movements that were consistent with the bouncing motions that are characteristic of superba during its cape presentation dances (Morrison-Scott, 1936). The displays of the captive also included apparent wing movement that produced “clicking” noises (Morrison-Scott, 1936). The only possibility of a distinct difference from superba that is evident in the descriptions of the captive minor is a very clear description of how the eye-spot ornaments have small black spots in their centers, which have the effect of looking like pupils (Morrison-Scott, 1936). It was hypothesized that the “pupils” were created through a concave surface to the feathers that create the eye-spots, which seems plausible. However, solid conclusions about the courtship phenotype of minor will have to await future field study.

Conclusions

Given the substantial differences in so many aspects of the niedda courtship phenotype combined with the acute geographic isolation of niedda populations from superba populations, leaves little doubt that niedda deserves full species status as proposed by Irestedt et al. (2017).

The species validity niedda underscores the importance of Indonesian New Guinea’ Bird’s Head and Bird’s Neck eco-regions (i.e., the Vogelkop Peninsula) as a center of endemic biodiversity that deserves particular attention from the conservation community. Among birds-of-paradise alone, L. niedda now joins the three other montane species endemic to the region (Astrapia nigra, Paradigalla longicuda, and Parotia sefilata) and opens the door for additional systematic scrutiny of the entire avifauna of the Bird’s Head (Vogeklop) region, including the other birds-of-paradise at middle and upper elevations (e.g., Drepanonris albertisi and Epimachus fastosus). Given that the Bird’s Head region was the first part of New Guinea to be ornithologically explored, and discovery of new species of birds-of-paradise was the main driver of exploration, it is surprising that the distinctive features of the niedda courtship phenotype, and therefore the species status of this unique population, have remained elusive for so long. Yet this fact underscores the need for continued exploration of New Guinea’s forests and further systematic investigation of all taxa, including those like the avifauna which are often considered to be relatively well known.

Supplemental Information

Figure S1 Courtship phenotype of L. superba superba

(A) Advertisement display and (B) vocalizing. (C) Horizontal display. (D) Pointing display. (E) Side view of the cape presentation display. (F) Frontal view of the cape presentation display. At the level of detail analyzed here, we found no discernible differences between the courtship phenotypes of superba superba and superba latipennis. Image credit/source: (A–D) Edwin Scholes/ML458167, (E) Kimberly Bostwick/ML487258 and (F) Kimberly Bostwick/ML487259.

Click here for additional data file.

Table S1 Media and associated metadata used for audiovisual analysis

Descriptive metadata and source information for the audiovisual data used in the analysis.

Click here for additional data file.

For field support and logistics we thank Shita Prativi, Benny Mambrasar and Jordan Mencher of Magnificus Expeditions and Papua Konservasi dan Komunitas. We offer sincere thanks to Pak Desa and the people of Minggre Village for granting access to their land, for being generous hosts and for excellent assistance in the field. We thank the Provincial Government of West Papua and Professor Charlie Heatubun, Head of the Research and Development Agency, for supporting of research and conservation in West Papua. Brad Walker, Matt Medler, Karen Rodriguez and Rick Elliker from the Cornell Lab of Ornithology provided critical support with multimedia data archival in the Macaulay Library. Bruce Beehler and an anonymous reviewer made valuable comments, which greatly improved the manuscript.

Additional Information and Declarations

Competing Interests

Author Contributions

Data Availability

The authors declare there are no competing interests.

Edwin Scholes conceived and designed the experiments, performed the experiments, analyzed the data, contributed reagents/materials/analysis tools, prepared figures and/or tables, authored or reviewed drafts of the paper, approved the final draft.

Timothy G. Laman conceived and designed the experiments, performed the experiments, contributed reagents/materials/analysis tools, authored or reviewed drafts of the paper, approved the final draft.

The following information was supplied regarding data availability:

Macaulay Library at the Cornell Lab of Ornithology

Accession #: ITM89474.

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
