# Peer review of "Distinctive courtship phenotype of the Vogelkop Superb Bird-of-Paradise Lophorina niedda Mayr, 1930 confirms new species status"

_PeerJ, doi:10.7717/peerj.4621_

## Round 0.1 · original submission · Minor Revisions

The reviews are generally positive and with adequate revisions your paper should be acceptable.

·

Basic reporting

This is a well-written paper, well documented, and well-argued, with an interesting subject.

Experimental design

The fieldwork done to support the hypothesis--that the Bird's Head population of the Superb Bird of Paradise should be treated as distinct is well-supported by the data and the argument.

Validity of the findings

I support the findings and think the paper achieves this objective as well.

Some issues to consider during the revision:

Beehler & Pratt (2016) highlighted the distinctness of the vocalization of the Bird's Head population of the Superb. This is the first published suggestion the population should possibly be considered distinct.

On line 22 the paper states "western Bird's Head" when in fact the bird inhabits the eastern Bird's Head. The MS should be searched and "western Bird's head" should be corrected to eastern Bird's Head.

Reviewer 2 ·

Basic reporting

In this investigation, Scholes and Laman provide novel audiovisual data on the courtship phenotype of the Vogelkop Superb Bird-of Paradise Lophorina niedda, corroborating recent molecular phylogeographic results by Irestedt et al 2017, and leaving little doubt that this distinctive taxon merits full species recognition. Some avian systematists (myself included) felt that Irestedt et al 2017 were premature in elevating the Vogelkop Superb BOP to full species given their reliance on just ~900 bp of cyt b and limited geographic sampling in the Bird’s Head/Neck region of New Guinea, leaving open the possibility that niedda and superba lineages may not be reciprocally monophyletic. The distinctive courtship repertoire and vocalizations documented by Scholes and Laman confirms that L. niedda is also isolated from adjacent L. superba populations by its unique display elements and advertisement calls, strengthening the argument for species recognition of this taxon. In general, this study is well written with clear research aims outlined in the introduction, sound presentation of results and conclusions, and relevant literature citations.

My primary concerns with this study center on the limited sample sizes that form the basis of behavioral analyses and the use of un-archived, copyright protected video data, that while viewable online, is not available in the raw formats needed for independent frame-by-frame behavioral analysis. I don’t think either of these issues should prevent the publication of this study, but the authors need to do a better job documenting their data sources and identifying the potential biases/limitations associated with small sample sizes, given that the stated secondary goal of this study is to establish a more thorough characterization of courtship phenotype in the Lophorina species complex for future comparative study. For example, telling the reader to just do a search for L. superba in the ML archive (Lines 132-134) doesn’t pass muster in terms of documenting which “behavioral specimens” were used to form the basis of analysis and description of specific display characters. I provide several recommendations herein that will help mitigate these issues and make this study a more useful contribution for future comparative analyses.

Introduction
Lines 63-68: It would be useful to mention the environmental display context (horizontal tree fall in the understory) somewhere in this paragraph so that the reader doesn’t have to wait for it in the results/discussion.

Materials and Methods
Line 117: A table or appendix listing the ML catalog #’s of the 30 practice displays and 14 displays to females is needed to document and summarize which clips form the basis of analysis. The table should include taxon name, ML # or privately held archive title, locality (lat,long), date filmed, and original source (videographer or audio recordist). A column for which display activity/element the clip illustrates along with time code notes would be useful. It should also be noted if the same individual is featured in multiple clips, as it looks like just three males (2 niedda and 1 superba) form the majority of the analyses.

Experimental design

No Comment

Validity of the findings

Results
The following ML #’s need be double checked, as they do not correspond to Lophorina specimens. I presume these should be 458000-458003?
Line 197: ML48000 Wrong taxon.
Line 227: ML48002 Wrong taxon.
Line 228: ML48003 Wrong taxon.

Lines 366-370: In general, I find the descriptions of the displays (Advertisement, Horizontal, Pointing, and Cape Presentation) and associated figures sufficiently clear and detailed to diagnose the primary differences between niedda and superba, but I feel the differences in tempo of the “foot stomping” element during the Cape Presentation display may deserve a bit more attention, as this aspect of the niedda display really struck me as being distinct from all other L. superba and L. minor displays that I have observed previously. The stomping in niedda seems to be faster and more pronounced in the absence of the superba bounce. Perhaps the authors could provide some hard numbers and give rates for the cadence of these elements, given they may well vary among local sky-island populations and be informative in diagnosing taxa.

Discussion
Lines 442-477: This section summarizes the key differences in display elements nicely and provides a well-reasoned argument for full species recognition of L. niedda. That being said, a statement at the end of this section is needed highlighting the caveats of basing these analyses on limited sample sizes and relying heavily on a single individual (superba) from a small outlying sky-island population in the Adelbert Mountains and just two individuals (niedda) from the Arfaks. The Vogelkop is a composite terrain with montane bird populations often exhibiting differences between the Tamarau and Arfak ranges. As such, the authors should identify these limitations and use them as a call for further exploration beyond just the easily accessible Arfak ranges.

Conclusions
Line 509-514: The high level of endemism in BOPs of the Vogelkop suggests to me that the entire regional Bird’s Head avifauna (as apposed to just the BOPs) merits additional systematic scrutiny, and stating as much would further strengthen the case for attention from the conservation community that this area of endemism deserves.

---

## Round 0.2 · accepted · Accept

Thanks for addressing all the reviewers comments and for acknowledging their help in improving the manuscript.

#